# Improving Time Series Forecasting via Instance-aware Post-hoc Revision

**Zhiding Liu[1], Mingyue Cheng[1], Guanhao Zhao[1], Jiqian Yang[1], Qi Liu[1], Enhong Chen[1]***

[1]State Key Laboratory of Cognitive Intelligence,
University of Science and Technology of China
{zhiding,ghzhao0223,yangjq}@mail.ustc.edu.cn
{mycheng,qiliuql,cheneh}@ustc.edu.cn

## Abstract

Time series forecasting plays a vital role in various real-world applications and has attracted significant attention in recent decades. While recent methods have achieved remarkable accuracy by incorporating advanced inductive biases and training strategies, we observe that instance-level variations remain a significant challenge. These variations—stemming from distribution shifts, missing data, and long-tail patterns—often lead to suboptimal forecasts for specific instances, even when overall performance appears strong. To address this issue, we propose a model-agnostic framework, **PIR**, designed to enhance forecasting performance through **P**ost-forecasting **I**dentification and **R**evision. Specifically, PIR first identifies biased forecasting instances by estimating their accuracy. Based on this, the framework revises the forecasts using contextual information, including co-variates and historical time series, from both local and global perspectives in a post-processing fashion. Extensive experiments on real-world datasets with mainstream forecasting models demonstrate that PIR effectively mitigates instance-level errors and significantly improves forecasting reliability. Our code is available[2]

## 1 Introduction

Time series forecasting is a fundamental task in time series analysis, attracting considerable attention in recent years [40, 49]. Various applications have been facilitated by the advancement of forecasting, including traffic planning [17], stock market prediction [23], healthcare analytics [19], and weather forecasting [3, 59]. Recent years have witnessed significant efforts dedicated to this area, with deep learning-based approaches achieving remarkable success due to their powerful ability to capture both temporal [62, 63] and cross-channel dependencies [60, 30]. Furthermore, advanced inductive biases and training strategies have been introduced to address the non-stationary nature of time series data [20, 33] and to construct foundation models for time series forecasting [9, 32, 52].

Despite their satisfactory performance in overall evaluations, we emphasize that existing forecasting approaches often overlook inherent instance-level variations, which can arise from the long-tail distribution of numerical patterns in time series data and ultimately lead to forecasting failures in specific cases. Specifically, time series typically represent the numerical reflections of complex and dynamic real-world systems and are therefore prone to noise, sensor failures, and other anomalies during data collection. These issues can result in potential distribution shifts [26], missing values [42], or unforeseen anomalies [4]. Therefore, while most instances exhibit similar numerical patterns, a subset may present rare behaviors, and mainstream forecasting methods often struggle to effectively model these exceptional instances, leading to inaccurate or even unreliable forecasting outcomes.

---

*Enhong Chen is the corresponding author.
[2]https://github.com/icantnamemyself/PIR

39th Conference on Neural Information Processing Systems (NeurIPS 2025).

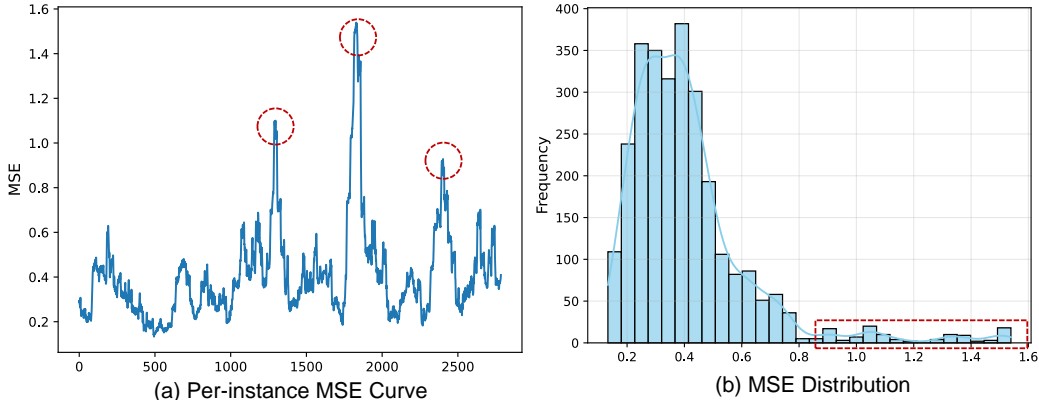

(a) Per-instance MSE Curve      (b) MSE Distribution

Figure 1: Per-instance MSE evaluation of PatchTST on the ETTh1 dataset and its corresponding error distribution. The error varies among instances and exhibits a long-tail distribution due to the instance-level variations.

To better illustrate this phenomenon, we present the per-instance Mean Squared Error (MSE) evaluation curve of PatchTST [37] on the ETTh1 dataset [62], along with the corresponding error distribution visualized through both a histogram and a kernel density estimate in Figure 1. As shown in the figure, while the MSE remains consistently low for the majority of instances, there exist specific cases where PatchTST yields unsatisfactory forecasting results, as indicated by multiple spikes in the error curve. Moreover, the error distribution clearly exhibits a long-tail pattern, reinforcing our motivation and underscoring the challenges posed by the instance-level variations.

To this end, we propose the **PIR** framework, designed to enhance forecasting results via **P**ost-forecasting **I**dentification and **R**evision, addressing the challenge from a novel post-processing perspective. The framework comprises two key components. The first is the failure identification mechanism, which identifies the potential biased forecasting instances where the model's predictions are less reliable, through estimating the forecasting performance on a per-instance level. The second is a post-revision module, which refines the forecasts by leveraging contextual information from both local and global perspectives. For the local revision, inspired by exogenous variable modeling approaches [38, 50], we utilize immediate forecasts of covariates along with available exogenous information such as timestamps [45] and textual descriptions [27] as side information to *implicitly* mitigate the impact of instance-level variations within a local window. This approach is grounded in the assumption that dependencies between covariates like lead-lag effects can provide valuable insights into future trends [61], and exogenous information can serve as additional prior conditions guiding the revision. For the global revision, the framework addresses rare or atypical numerical patterns by *explicitly retrieving* similar instances from the global historical data [28]. This retrieval-based strategy enables the model to better capture long-tail patterns that may be overlooked by conventional forecasting models. Finally, PIR integrates the original forecasts with its revision outputs via a weighted sum, making it model-agnostic and broadly compatible with existing forecasting architectures. The primary contributions of our work are summarized as follows:

- We are the first to highlight the existence of instance-level variations that can lead to forecasting failures on certain cases, evident by the unsatisfactory performance of existing forecasting methods at the per-instance level.

- We introduce PIR, a model-agnostic framework designed to address this challenge. The framework estimates the forecasting performance to identify the potential failures and utilizes contextual information to revise them from both local and global perspectives.

- We conduct extensive experiments on well-established real-world datasets, covering both long-term and short-term forecasting settings with mainstream models. The results demonstrate that the PIR framework consistently enhances forecasting accuracy, leading to more reliable and robust performance.

## 2    Related Work

### 2.1    Time Series Forecasting

Time series forecasting has been a central area of research for several decades. One of the earliest landmark contributions was the development of statistical methods, which are celebrated for their solid theoretical foundation and systematic approach to model design. Representative works in this domain include ARIMA and Holt-Winters [5, 18], which have laid the groundwork for more advanced methodologies. More recently, with the advancement of deep learning, numerous neural forecasting models have been proposed, achieving superior performance. Recurrent Neural Networks (RNNs) are among the first deep learning architectures applied to forecasting [51, 41], followed by a variety of other structures, such as convolution-based models [21, 35, 7], attention-based networks [22, 24, 63], and MLP-based architectures [39, 57, 25]. These models have demonstrated remarkable capabilities in capturing both temporal and cross-channel dependencies within time series data, leading to substantial improvements in forecasting accuracy.

Beyond advancements in model architecture, a range of specialized techniques rooted in time series analysis has also emerged. These include methods for time series decomposition [54, 56], frequency modeling [55], and approaches for non-stationary forecasting [20, 33], which address the unique challenges presented by time series data. Besides, self-supervised pretraining has gained significant attention as a powerful approach, with various training strategies being explored [53, 6, 58]. In addition, recent efforts aim to construct foundational time series models capable of learning universal representations and being applied to diverse downstream tasks [34, 9, 32, 52]. Moreover, some pioneer works explore forecasting enhanced with the reasoning ability of LLMs [46, 36].

### 2.2    Context Modeling in Time Series Forecasting

In addition to forecasting based solely on multivariate time series data, several pioneering studies have explored the incorporation of contextual information to enhance forecasting performance [1]. On one hand, some advancements focus on integrating exogenous information within a local window during the forecasting process. For instance, TFT [24] and TiDE [8] leverage dense encoders to process timestamp information, which is subsequently used to condition future forecastings. Similarly, NBEATSx [38] and TimeXer [50] improves the forecasting accuracy of target variables by explicitly modeling the influence of exogenous variables. On the other hand, some studies investigate the potential of retrieving relevant time series from the global historical context. RATD [28] utilizes the retrieved series as references to guide the denoising process of diffusion-based forecasters, and RATSF [48] introduces the retrieval augmented cross-attention architecture for explicitly modeling similar historical data. More recently, RAFT [14] achieves satisfactory performance through forecasting enhanced with multi-period retrieval.

Different from existing methods, our proposed PIR framework tackles a novel research challenge: mitigating the impact of instance-level variations that can lead to forecasting failures in certain cases. The framework begins by identifying the potential failure instances through estimating their accuracy, and then revises the forecasts by leveraging available contextual information from both local and global perspectives in a post-processing manner. As a result, PIR functions as a model-agnostic plugin, allowing it to be seamlessly integrated into arbitrary forecasting models for enhanced performance.

## 3    Methodology

In this section, we will delve into the specifics of the proposed PIR framework, which is illustrated in Figure 2. We begin with the problem definition, followed by a comprehensive description of the framework's key components.

### 3.1    Problem Definition

We first consider the general multivariate time series forecasting task. Given a set of input series $X = \{x_i\}_{i=1}^{M}$, the objective is to learn a mapping function $Y = f_\theta(X)$ that accurately forecasts the future values $Y = \{y_i\}_{i=1}^{M}$, where $x_i \in \mathbb{R}^{N \times L_{in}}$ represents the $i$-th input time series and

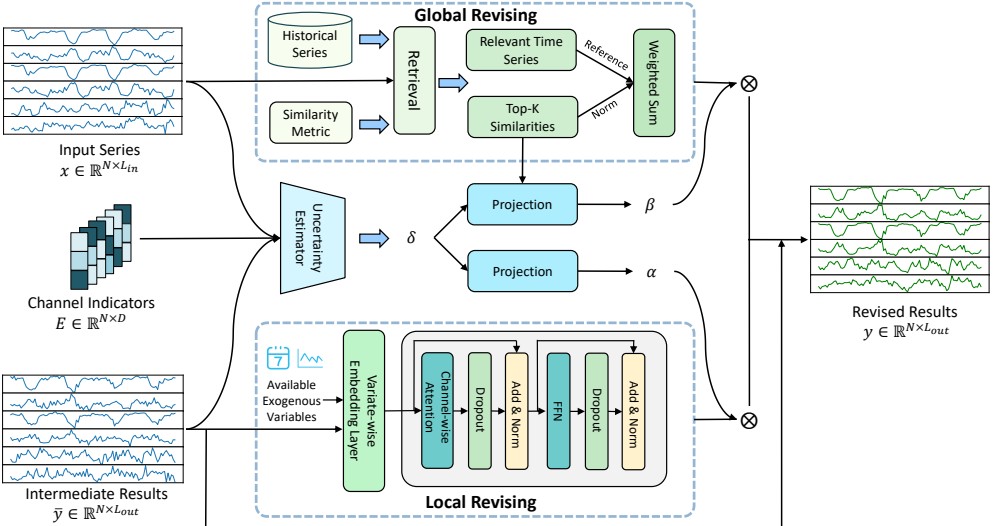

Figure 2: Overview of the proposed PIR framework. The framework first identifies the potential forecasting failure cases through estimating the performance of the intermediate results generated by backbone models on a per-instance level. Besides, the framework incorporates Local Revising and Global Revising components, which utilize contextual information, including available exogenous variables within the local window and global historical time series, to revise the forecasting results for enhanced performance.

$y_i \in \mathbb{R}^{N \times L_{out}}$ represents the corresponding target series. Here, $N$ denotes the number of channels and $M$ is the number of instances, while $L_{in}$ and $L_{out}$ denote the lengths of the input and target series, respectively. For simplicity and brevity, we will omit the indices for the time series instances in the following sections.

Moreover, the PIR framework focuses on a novel problem to revise the forecasting results for better performance. Specifically, given the intermediate results $\bar{Y}$ of arbitrary forecasting models, the objective is to learn the revising function $Y = f_\phi(X, \bar{Y}, C)$ that corrects the potential forecasting failures. Here, $C$ represents the available contextual information, such as exogenous variables, that can aid in the revision process for more accurate and reliable forecasts.

### 3.2 Failure Identification

The goal of the PIR framework is to post-process forecasting results by identifying and revising potential forecasting failures caused by instance-level variations. Thus, the identification process becomes a primary target and challenge. Conceptually, it forms an *Uncertainty Estimation* task based on the immediate forecasting results. Though various efforts have been devoted in this area, it has not been deeply explored for properly estimating the uncertainty of point-wise forecasting results mainly due to two reasons. Firstly, given the regression nature of the forecasting task, most existing approaches generate predictions directly from the hidden states. As a result, token-level distributions before generation are not available, and common uncertainty evaluation methods based on probabilities are not directly applicable [2, 11]. Secondly, the forecasting failures arise from both data uncertainty (e.g., missing values) and model uncertainty (e.g., underfitting on long-tail patterns) [12] while most existing uncertainty quantification methods primarily focus on the latter one.

Although we have identified several potential reasons that may lead to prediction failure, there are many other potential and interrelated factors, and there is a lack of ground truth and metrics to identify and disentangle the specific reasons for each forecasting failure instance. Therefore, we do not explicitly locate these failure patterns. Instead, we innovatively *quantify uncertainty using forecasting error* and adopt a data-driven approach as a practical solution for estimating the uncertainty of a given input-forecast pair $(x, \bar{y})$, which is more flexible and can potentially accommodate a broader

range of failure modes. Specifically, following common practice in modeling complex dynamics [31, 10], we employ a two-layer fully connected neural network with non-linear activation functions $f_{ue}(\cdot)$ to estimate the forecasting uncertainty. Additionally, we introduce an auxiliary constraint to support the abovementioned procedure. Since explicit uncertainty is not directly available for the forecasting results, we treat the forecasting error as a feasible guiding signal. The intuition behind this is that higher uncertainty will likely result in greater forecasting error. Therefore, the estimated uncertainty is further constrained to predict the MSE of the forecasting results:

$$\delta = f_{ue}(x, \bar{y}, E),$$
$$\mathcal{L}_{ue} = \frac{1}{N} \sum_1^N \|\delta - \|\bar{y} - y\|_2^2\|_1. \tag{1}$$

Here $\delta$ is the estimated uncertainty, and $\|\cdot\|_1$ represents the Mean Absolute Error (MAE) loss function, which ensures that the estimated uncertainty aligns with the actual forecasting error. To provide additional contextual information, we introduce the channel embedding matrix $E = (e_1, e_2, ..., e_N) \in \mathbb{R}^{N \times d}$, which encodes the channel identity information. This matrix enables the model to better capture variations across different channels and provides crucial context for more accurate uncertainty estimation. Through this approach, the framework can generate uncertainty estimates specific to each model and input instance, thereby satisfying both types of uncertainty. Moreover, it forms a coupled framework, enabling efficient end-to-end optimization jointly with the forecasting task, and offers a degree of interpretability by using the forecasting error as a measurement for uncertainty.

### 3.3 Local Revising

After identifying the potential forecasting failures with high uncertainty, the PIR framework revises these results from both local and global perspectives. For the local revision component, the core idea is to leverage the contextual information within a local horizon window, which includes the intermediate forecastings of covariates and available exogenous variables, to enhance forecasting accuracy. The intuition behind this approach is straightforward. Firstly, the dependencies including lead-lag effects are widely present in the time series data, where the forecasting results of covariates can provide valuable insights into future trends [61]. Besides, the exogenous variables, such as the time-related information and environmental factors known in advance can serve as a prior condition [24, 45], mitigating the impact of sudden distribution shifts caused by natural rhythms like holiday effects [15, 43]. Conceptually, this approach is particularly beneficial for models that prioritize robustness over capacity by adopting a channel-independent strategy [13].

In practice, we project the intermediate forecasting results into hidden states on a per-variate basis [30], along with the available exogenous variable information, which are concatenated for further correlation extraction. Let $h_{co}, h_{exo}$ be the representation of covariates and exogenous variables respectively, and $c$ refer to the available exogenous variable information corresponding to the instance, the projection process is formulated as:

$$H_0 = [h_{co}, h_{exo}],$$
$$h_{co} = \text{CoVariateEmb}(\bar{y}), \tag{2}$$
$$h_{exo} = \text{ExoVariateEmb}(c).$$

Here CoVariateEmb$(\cdot)$ is a trainable linear projector, and ExoVariateEmb$(\cdot)$ is implemented flexibly based on the characteristic of $c$, which can be a linear projector for numerical features or language models for textual descriptions [27]. Through this approach, local context from multiple sources and domains can be seamlessly embedded in the framework to guide the revising process.

The hidden states are then passed through a traditional Transformer [44] with a linear prediction head to generate the revised results $y_{local}$. By leveraging the attention mechanism, the local revision component explicitly captures the correlations between covariates and exogenous variables, thereby ensuring that the contextual information within the local window is fully utilized to correct forecasting failures and enhance prediction accuracy.

### 3.4 Global Revising

In addition, the PIR framework incorporates a global revision component that leverages global historical information to further refine forecasting results. As illustrated in Figure 1, instance-level

variances can lead to a long-tail distribution in performance, primarily because traditional models struggle to capture rare numerical patterns. To address this challenge, we introduce a straightforward yet effective retrieval module that explicitly retrieves relevant historical time series exhibiting similar numerical characteristics. These retrieved series serve as reference signals during the revision process, enabling the model to better handle atypical patterns that are typically underrepresented in training.

Specifically, we first construct the retrieval database using only the training input-target time series pairs $(X_{train}, Y_{train})$, as we treat historical information as the global context. This design not only prevents data leakage but also facilitates the extension of the database to incorporate multi-source datasets, since it depends solely on raw time series data. Based on the retrieval database, the top-$K$ most relevant time series are selected for a given input series $x$ by computing the similarity between $x$ and each candidate in the database, as formalized below:

$$\text{Index}, w = \text{TopK } \text{Sim}(Enc(x), Enc(X)),$$
$$Y_{re} = \{Y_{train,i} | \forall i \in \text{Index}\}. \tag{3}$$

Here, $Enc(\cdot)$ is an encoding function that processes the time series, which is instantiated as an instance normalization operation [20] in practice for its simplicity and effectiveness of mitigating the impact of non-stationarity that could lead to unstable similarity estimation. Moreover, powerful pre-trained forecasting models [9, 32, 52] can be optionally utilized for better representation projection. Additionally, $w$ represents the top-$K$ similarity scores estimated by the similarity operator $\text{Sim}(\cdot, \cdot)$, which is instantiated using cosine similarity due to its computational efficiency.

Once the relevant time series are retrieved, they serve as references for revising the forecasting results. Instead of modifying the architecture of the backbone forecasting models to incorporate these references, we adopt a practical assumption: similar instances tend to exhibit similar future trends. Therefore, the retrieved references themselves can serve as effective estimations for the target series. This design ensures that the framework remains entirely independent of the backbone models, making it applicable to any forecasting model. In practice, the similarity scores are treated as importance indicators, and the global revised results $y_{global}$ are generated through a weighted sum operation, formulated as follows:

$$p = \text{Softmax}(w),$$
$$y_{global} = \text{WeightedSum}(p, Y_{re}), \tag{4}$$

where the $\text{Softmax}(\cdot)$ function ensures that $\sum_{i=1}^{K} p_i = 1$, assigning higher weights to retrieved instances that are more similar to the input series.

### 3.5 Optimization Target

By combining the components described above, the PIR framework produces the final forecasting results through post-forecasting identification and revision using a residual approach [16]:

$$y_{pred} = \bar{y} + \alpha y_{local} + \beta y_{global},$$
$$\alpha = \sigma(\text{Linear}(\delta)), \tag{5}$$
$$\beta = \sigma(\text{MLP}(\delta, w)).$$

Here, $\alpha$ and $\beta$ are learned weights corresponding to the local and global revision components, respectively, and $\sigma(\cdot)$ denotes the Sigmoid activation function. A linear transformation is employed to estimate $\alpha$, with its weight and bias initialized to vectors of ones and zeros. This design ensures ensuring a positive correlation that higher uncertainty estimates lead to larger values of $\alpha$, thus placing greater emphasis on the local revision. In contrast, $\beta$ is generated through a multi-layer perceptron (MLP), which takes both the estimated uncertainty and the retrieval similarities as input. This allows the model to dynamically adjust the influence of the global revision based on the confidence of the prediction and the relevance of the retrieved historical series. The overall optimization objective is to minimize the MSE between the revised forecasts and the ground truth, formulated as:

$$\mathcal{L}_{pr} = \frac{1}{N} \sum_{1}^{N} \|y_{pred} - y\|_2^2. \tag{6}$$

Finally, let $\lambda$ denote the weight hyperparameters for the auxiliary constraint defined in Section 3.2, the overall optimization objective for the PIR framework is then formulated as a multitask learning problem:

$$\mathcal{L} = \mathcal{L}_{pr} + \lambda \mathcal{L}_{ue}. \tag{7}$$

# 4 Experiments

In this section, we conduct extensive experiments on widely used benchmark datasets, comparing our proposed PIR framework with mainstream forecasting approaches under both long-term and short-term forecasting settings, to demonstrate its effectiveness.

Table 1: Forecasting performance under long-term and short-term settings. The results are averaged from all the target series lengths, and the full results are provided in the Appendix. The **bold** values indicate better performance.

| Methods | | PatchTST | + PIR | Imp.(%) | SparseTSF | + PIR | Imp.(%) | iTrans. | + PIR | Imp.(%) | TimeMixer | + PIR | Imp.(%) |
|---|---|---|---|---|---|---|---|---|---|---|---|---|---|
| ETTh1 | MSE | 0.466 | **0.437** | 6.22 | 0.444 | **0.433** | 2.48 | 0.451 | **0.432** | 4.21 | 0.445 | **0.429** | 3.60 |
| | MAE | 0.452 | **0.439** | 2.88 | 0.431 | **0.429** | 0.46 | 0.445 | **0.437** | 1.80 | 0.436 | **0.431** | 1.15 |
| ETTh2 | MSE | 0.384 | **0.375** | 2.34 | 0.384 | **0.373** | 2.86 | 0.383 | **0.377** | 1.57 | 0.380 | **0.378** | 0.53 |
| | MAE | 0.405 | **0.400** | 1.23 | 0.403 | **0.398** | 1.24 | 0.406 | **0.403** | 0.74 | 0.405 | **0.403** | 0.49 |
| ETTm1 | MSE | 0.397 | **0.383** | 3.53 | 0.416 | **0.378** | 9.13 | 0.407 | **0.383** | 5.90 | 0.381 | **0.377** | 1.05 |
| | MAE | 0.408 | **0.397** | 2.70 | 0.407 | **0.390** | 4.18 | 0.412 | **0.397** | 3.64 | 0.396 | **0.393** | 0.76 |
| ETTm2 | MSE | **0.281** | 0.283 | -0.71 | 0.287 | **0.281** | 2.09 | 0.291 | **0.288** | 1.03 | 0.276 | **0.274** | 0.72 |
| | MAE | **0.329** | 0.330 | -0.30 | 0.329 | **0.328** | 0.30 | **0.334** | **0.334** | 0.00 | **0.322** | 0.323 | -0.31 |
| Electricity | MSE | 0.215 | **0.200** | 6.98 | 0.224 | **0.196** | 12.50 | 0.179 | **0.175** | 2.23 | 0.185 | **0.181** | 2.16 |
| | MAE | 0.303 | **0.279** | 7.92 | 0.297 | **0.275** | 7.41 | 0.269 | **0.265** | 1.49 | 0.275 | **0.270** | 1.82 |
| Solar | MSE | 0.269 | **0.244** | 9.29 | 0.385 | **0.275** | 28.57 | 0.236 | **0.231** | 2.12 | 0.231 | **0.223** | 3.46 |
| | MAE | 0.307 | **0.287** | 6.51 | 0.370 | **0.296** | 20.00 | 0.263 | **0.260** | 1.14 | 0.270 | **0.267** | 1.11 |
| Weather | MSE | 0.259 | **0.254** | 1.93 | 0.276 | **0.261** | 5.43 | 0.260 | **0.255** | 1.92 | 0.245 | **0.244** | 0.41 |
| | MAE | 0.281 | **0.277** | 1.42 | 0.294 | **0.282** | 4.08 | 0.280 | **0.277** | 1.07 | **0.274** | **0.274** | 0.00 |
| Traffic | MSE | 0.482 | **0.459** | 4.77 | 0.637 | **0.477** | 25.12 | 0.425 | **0.420** | 1.18 | 0.519 | **0.492** | 5.20 |
| | MAE | 0.308 | **0.299** | 2.92 | 0.379 | **0.314** | 17.15 | 0.283 | **0.280** | 1.06 | 0.307 | **0.291** | 5.21 |
| PEMS03 | MSE | 0.158 | **0.120** | 24.05 | 0.351 | **0.154** | 56.13 | 0.115 | **0.107** | 6.96 | **0.089** | **0.089** | 0.00 |
| | MAE | 0.265 | **0.231** | 12.83 | 0.400 | **0.261** | 34.75 | 0.225 | **0.216** | 4.00 | **0.200** | **0.200** | 0.00 |
| PEMS04 | MSE | 0.206 | **0.140** | 32.04 | 0.370 | **0.171** | 53.78 | 0.108 | **0.097** | 10.19 | 0.083 | **0.081** | 2.41 |
| | MAE | 0.305 | **0.251** | 17.70 | 0.419 | **0.278** | 33.65 | 0.220 | **0.206** | 6.36 | 0.191 | **0.190** | 0.52 |
| PEMS07 | MSE | 0.165 | **0.115** | 30.30 | 0.352 | **0.149** | 57.67 | 0.095 | **0.089** | 6.32 | 0.087 | **0.083** | 4.60 |
| | MAE | 0.268 | **0.223** | 16.79 | 0.404 | **0.249** | 38.37 | 0.198 | **0.189** | 4.55 | 0.191 | **0.187** | 2.09 |
| PEMS08 | MSE | 0.186 | **0.147** | 20.97 | 0.362 | **0.180** | 50.28 | 0.147 | **0.135** | 8.16 | 0.130 | **0.128** | 1.54 |
| | MAE | 0.284 | **0.251** | 11.62 | 0.413 | **0.279** | 32.45 | 0.245 | **0.237** | 3.27 | 0.238 | **0.232** | 2.52 |

## 4.1 Experimental Setup

**Datasets.** For the long-term forecasting task, we conduct experiments on a widely recognized benchmark dataset that includes eight real-world datasets spanning diverse domains [54, 21]. Additionally, we incorporate the PEMS dataset, which contains four subsets, for the short-term forecasting task [29]. The exogenous information used in these datasets are the available timestamps. We also conduct experiments on datasets with additional textual descriptions in the Appendix. Following standard experimental protocols, we split each dataset into training, validation, and testing sets in chronological order. The split ratios are set to 6:2:2 for the ETT dataset and 7:1:2 for the other datasets. Detailed information about the datasets is available in the Appendix.

**Backbone Models.** PIR is a model-agnostic plugin that can be seamlessly integrated with any time series forecasting model. To demonstrate the effectiveness of the framework, we select four mainstream forecasting models based on diverse architectures, encompassing both channel-dependent and channel-independent assumptions, as backbones. These models are evaluated under both long-term and short-term forecasting settings: **PatchTST** [37], **SparseTSF** [25], **iTransformer** [30] and **TimeMixer** [47]. We implement these models following their official code.

**Experiments Details.** We employ the ADAM optimizer as the default optimization algorithm across all experiments and evaluate performance using two metrics: mean squared error (MSE) and mean absolute error (MAE). For the PIR framework, the retrieval number $K$ is tuned from the set $\{10, 20, 50\}$, and the weight hyperparameter $\lambda$ is fixed at 1. All experiments are conducted on a single NVIDIA RTX 4090 GPU.

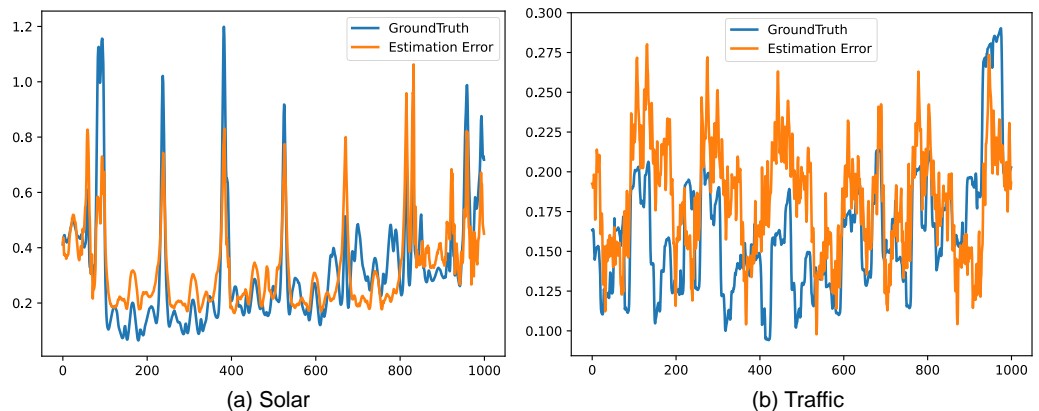

Figure 3: Comparison of the groundtruth forecasting error and the estimation of PIR on Solar and Traffic dataset. The backbone is SparseTSF, and the target length is set to 336.

## 4.2 Main Results

Following the standard evaluation protocol [62, 56], we set the input series length $L_{in} = 96$ across all datasets. For a unified comparison, the target series length $L_{out}$ is set to $\{12, 24, 36, 48\}$ for the PEMS dataset and $\{96, 192, 336, 720\}$ for the remaining datasets. The forecasting results for both long-term and short-term settings, along with the corresponding relative performance improvements, are summarized in Table 1.

As shown in the table, the proposed PIR framework consistently improves the performance of state-of-the-art forecasting models across most scenarios in both long-term and short-term forecasting settings. This improvement is primarily due to PIR's capability to identify potential forecasting failures and effectively leverage contextual information for revision. Specifically, across all 48 experimental settings, PIR achieves an average MSE reduction of **8.99%** for PatchTST. Similar trends are observed for other backbone models, with reductions of **25.87%** for SparseTSF, **3.47%** for iTransformer, and **2.34%** for TimeMixer. These results underscore the generalizability of the PIR framework, demonstrating its seamless integration with diverse forecasting models, regardless of their architecture or inductive biases, to deliver enhanced predictive performance.

Additionally, we observe that the relative performance improvements for channel-dependent approaches are smaller compared to those for channel-independent models. This is likely because channel-dependent methods already incorporate covariate information, resulting in stronger baseline performance and leaving less room for improvement. Nevertheless, by leveraging the forecasted covariates, future exogenous variables, and historical time series as contextual information, the PIR framework still manages to enhance the performance of these models. Notably, although the local revision component of PIR shares a structural resemblance with iTransformer, it continues to yield substantial relative improvements. These findings further support our hypothesis that instance-level variations contribute to forecasting failures in specific cases. Moreover, the comparison also highlights the importance of post-forecasting failure identification and revision in generating more reliable and robust forecasting results.

## 4.3 Qualitative Analysis

To better illustrate how the PIR framework works to help enhance the forecasting performance, we provide a qualitative analysis of the identification and revision process in this section.

We begin by evaluating the reliability of the failure identification component through a comparison between the actual forecasting error of the backbone models and PIR's estimated error, $\delta$, as defined in Equation 1. Specifically, we visualize both metrics over a segment of the test set from the Solar and Traffic datasets, using SparseTSF as the backbone model, in Figure 3. The results demonstrate that the PIR framework accurately estimates the forecasting error of the backbone's intermediate predictions.

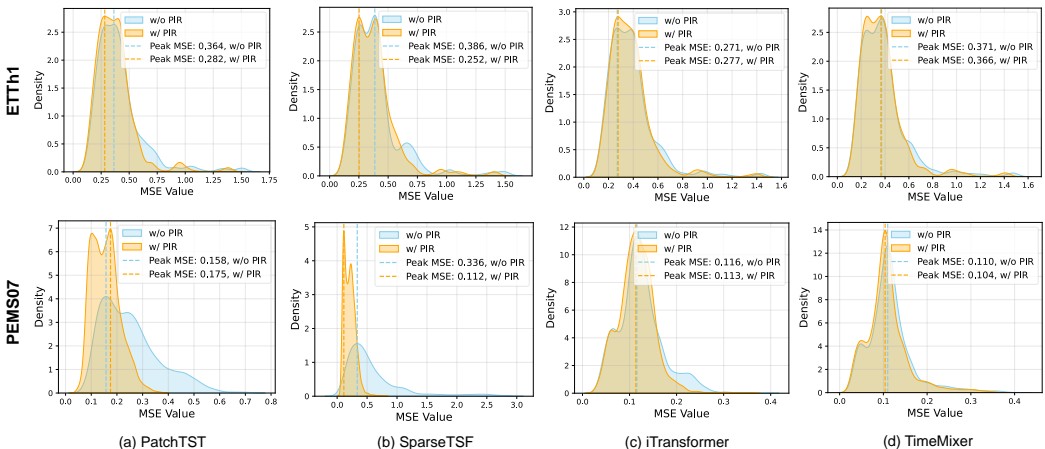

Figure 4: Illustration of the per-instance MSE distribution differences for four mainstream forecasting approaches, both with and without the enhancement of PIR. The MSE values corresponding to the peak density are also highlighted. The comparisons are performed on the ETTh1 and PEMS07 datasets, with the target length set to 96 and 48.

Notably, the estimated and actual error curves exhibit consistent patterns in terms of peaks and troughs, which validates PIR's capability to assess the quality of individual forecasting results. This strong alignment underscores the effectiveness of PIR's uncertainty estimation mechanism and its ability to reliably identify potential forecasting failures, laying a solid foundation for the subsequent revision stage.

For the revision component, although the quantitative results reported in Table 1 already demonstrate its effectiveness in improving overall forecasting performance, we further investigate how the PIR framework mitigates the effects of instance-level variance. Specifically, we evaluate the per-instance MSE of the baseline backbone models on the ETTh1 and PEMS07 datasets. As illustrated in Figure 4, we compare the MSE distribution with and without PIR enhancement, measured using kernel density estimation, and highlight the MSE corresponding to the peak density for each case. The figure reveals that models enhanced by PIR yield more reliable forecasting results, as their error distributions exhibit higher density at lower MSE values. Additionally, the MSE corresponding to the peak density is significantly reduced. Additionally, PIR framework also effectively addresses forecasting failures, substantially reducing errors for instances poorly modeled by the backbone models. This improvement is particularly evident in the PEMS07 dataset, where the tail of the error distribution curve shifts significantly toward smaller MSE values. For example, the maximum prediction error of SparseTSF on ETTh1 decreases from 2.85 to 0.81 when enhanced with PIR.

In summary, the qualitative analysis strongly aligns with our expectations. The PIR framework can effectively identify forecasting failures and enhance overall performance by revising the intermediate results in a model-agnostic manner.

## 4.4 Complexity Analysis

In this section, we analyze the computational overhead of the proposed PIR framework from both theoretical and empirical perspectives. Theoretically, the series-wise cosine similarity function used in the retrieval stage has a time complexity of $O(NML_{in})$, where $N$ denotes the number of channels, $M$ is the total number of historical series per channel, and $L_{in}$ represents the input sequence length. The subsequent local revision process involves channel-wise attention operations, resulting in a complexity of $O(N^2)$. To further evaluate the practical efficiency of PIR, we report the average inference time on both a small-scale dataset (ETTh1) and a large-scale dataset (Traffic). We compare two retrieval strategies: brute-force cosine similarity and approximate retrieval using Locality-Sensitive Hashing (LSH). The results are summarized in Table 2.

Table 2: Inference time comparison under different settings.

|  | ETTh1(Cos) | ETTh1(LSH) | Traffic(Cos) | Traffic(LSH) |
|---|---|---|---|---|
| Backone(s) | 0.164 | 0.164 | 0.424 | 0.424 |
| $\Delta$retrieval(s) | 0.024 | 0.415 | 0.079 | 87.957 |
| $\Delta$revision(s) | 0.096 | 0.096 | 0.275 | 0.275 |
| MSE Improvement | 0.014 | 0.009 | 0.025 | 0.025 |

The results indicate that the retrieval stage introduces negligible additional latency on both datasets, thanks to the GPU-parallelizable nature of cosine similarity. For even larger datasets, the total computational cost can be further reduced by applying sampling strategies (e.g., stride sampling) or dimensionality reduction techniques to limit the search space. In contrast, the LSH-based retrieval implemented with the `faiss` library yields significantly higher inference time without performance gains, indicating that brute-force cosine similarity is both more efficient and effective in our current implementation.

## 5 Conclusion

In this paper, we first investigated the challenge of instance-level variance in time series forecasting, which often results in unreliable predictions for certain cases. To address this, we proposed the PIR framework, a model-agnostic solution that enhances the accuracy through post-forecasting identification and revision of potentially biased predictions. The framework identifies the forecasting failures by estimating their error on a per-instance basis, and leverages contextual information to revise forecasts from both local and global perspectives. Extensive experiments across various benchmarks and forecasting models demonstrated that PIR consistently improves performance, highlighting its effectiveness and versatility as a plug-in component for a wide range of forecasting architectures. We wish our work could raise new research directions for the forecasting task.

## 6 Acknowledgement

This work was supported by the National Natural Science Foundation of China (No. U23A20319, No. 62502486). We furthermore thanked the anonymous reviewers for their constructive comments.

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

# A    Dataset Descriptions

In this paper, we leverage a diverse set of forecasting datasets covering various domains to evaluate the effectiveness of the proposed PIR framework under both long-term and short-term forecasting settings. We also include datasets with additional textual information [27] to validate the framework's generalizability. The brief descriptions and characteristics are presented as follows:

- **ETT**[3]**:** The dataset records oil temperature and load metrics from electricity transformers, tracked between July 2016 and July 2018. It is subdivided into four mini-datasets, with data sampled either hourly or every 15 minutes.
- **Electricity**[4]**:** The dataset captures the hourly electricity consumption in kWh of 321 clients, monitored from July 2016 to July 2019.
- **Solar**[5]**:** The dataset records the solar power production in the year of 2006, which is sampled every 10 minutes from 137 PV plants in Alabama State.
- **Weather**[6]**:** The dataset records the 21 weather indicators, including air temperature and humidity every 10 minutes from the Weather Station of the Max Planck Biogeochemistry Institute in 2020.
- **Traffic**[7]**:** The dataset provides the hourly traffic volume data describing the road occupancy rates of San Francisco freeways, recorded by 862 sensors.
- **PEMS**[8]**:** The dataset is a series of traffic flow dataset with four subsets, including PEMS03, PEMS04, PEMS07, and PEMS08. The traffic information is recorded at a rate of every 5 minutes by multiple sensors.
- **Energy** and **Health**[9]**:** These two datasets are subsets of Time-MMD [27], a multimodal time series dataset that ensures fine-grained alignment between textual and numerical modalities. The datasets are collected weekly, spanning from 1996 and 1997 up to May 2024, respectively.

Table 3: The overview of each dataset used in the experiments.

| Dataset | Variables | Frequency | Length | Scope |
|---------|-----------|-----------|--------|-------|
| ETTh1&ETTh2 | 7 | 1 Hour | 17420 | Energy |
| ETTm1&ETTm2 | 7 | 15 Minutes | 69680 | Energy |
| Electricity | 321 | 1 Hour | 26304 | Energy |
| Solar | 137 | 10 Minutes | 52560 | Nature |
| Weather | 21 | 10 Minutes | 52696 | Nature |
| Traffic | 862 | 1 Hour | 17544 | Transportation |
| PEMS03 | 358 | 5 Minutes | 26208 | Transportation |
| PEMS04 | 307 | 5 Minutes | 16992 | Transportation |
| PEMS07 | 883 | 5 Minutes | 28224 | Transportation |
| PEMS08 | 170 | 5 Minutes | 17856 | Transportation |
| Energy | 9 | 1 Week | 1479 | Energy |
| Health | 11 | 1 Week | 1389 | Health |

# B    Full Experimental Results

In this section, we present the complete long-term and short-term forecasting results in Table 4. These results demonstrate that the proposed PIR framework functions as a model-agnostic plugin,

---

[3]https://github.com/zhouhaoyi/ETDataset

[4]https://archive.ics.uci.edu/ml/datasets/ElectricityLoadDiagrams20112014

[5]http://www.nrel.gov/grid/solar-power-data.html

[6]https://www.bgc-jena.mpg.de/wetter/

[7]http://pems.dot.ca.gov

[8]https://github.com/guoshnBJTU/ASTGNN/tree/main/data

[9]https://github.com/AdityaLab/MM-TSFlib

Table 4: Full experimental results of long-term and short-term forecasting. The target length $L_{out}$ is chosen as {12,24,36,48} for the PEMS dataset and {96,192,336,720} for the others. The **bold** values indicate better performance.

| Methods Metric | PatchTST MSE | PatchTST MAE | + PIR MSE | + PIR MAE | SparseTSF MSE | SparseTSF MAE | + PIR MSE | + PIR MAE | iTransformer MSE | iTransformer MAE | + PIR MSE | + PIR MAE | TimeMixer MSE | TimeMixer MAE | + PIR MSE | + PIR MAE |
|---|---|---|---|---|---|---|---|---|---|---|---|---|---|---|---|---|
| **ETTh1** 96 | 0.410 | 0.416 | **0.375** | **0.400** | 0.399 | 0.401 | **0.375** | **0.392** | 0.385 | 0.403 | **0.376** | **0.402** | 0.384 | 0.399 | **0.370** | **0.397** |
| 192 | 0.458 | 0.443 | **0.422** | **0.427** | 0.438 | 0.423 | **0.420** | **0.416** | 0.440 | 0.434 | **0.424** | **0.429** | 0.432 | 0.425 | **0.422** | **0.421** |
| 336 | 0.498 | 0.464 | **0.467** | **0.451** | 0.478 | 0.443 | **0.465** | **0.440** | 0.486 | 0.457 | **0.465** | **0.450** | 0.486 | 0.449 | **0.460** | **0.442** |
| 720 | 0.498 | 0.486 | **0.484** | **0.477** | **0.461** | **0.457** | 0.473 | 0.466 | 0.494 | 0.485 | **0.461** | **0.466** | 0.476 | 0.470 | **0.464** | **0.462** |
| **ETTh2** 96 | 0.299 | 0.347 | **0.291** | **0.340** | 0.304 | 0.347 | **0.289** | **0.338** | 0.302 | 0.350 | **0.298** | **0.346** | 0.295 | **0.343** | **0.291** | 0.343 |
| 192 | 0.384 | 0.398 | **0.371** | **0.391** | 0.386 | 0.396 | **0.373** | **0.392** | 0.382 | 0.399 | **0.374** | **0.396** | 0.378 | 0.398 | **0.367** | **0.395** |
| 336 | 0.424 | 0.432 | **0.418** | **0.429** | 0.424 | 0.429 | **0.418** | **0.426** | 0.423 | 0.433 | **0.415** | **0.429** | 0.421 | 0.436 | 0.422 | 0.431 |
| 720 | 0.427 | 0.444 | **0.421** | **0.441** | 0.421 | 0.438 | **0.412** | **0.435** | 0.424 | 0.443 | **0.420** | **0.440** | 0.427 | 0.441 | 0.430 | 0.444 |
| **ETTm1** 96 | 0.336 | 0.375 | **0.317** | **0.354** | 0.357 | 0.375 | **0.311** | **0.350** | 0.342 | 0.377 | **0.315** | **0.356** | 0.316 | 0.355 | **0.309** | **0.351** |
| 192 | 0.376 | 0.394 | **0.365** | **0.383** | 0.394 | 0.392 | **0.355** | **0.374** | 0.381 | 0.395 | **0.358** | **0.380** | 0.364 | 0.383 | **0.359** | **0.381** |
| 336 | 0.409 | 0.416 | **0.394** | **0.404** | 0.426 | 0.414 | **0.389** | **0.396** | 0.420 | 0.420 | **0.395** | **0.406** | 0.389 | 0.403 | **0.387** | **0.402** |
| 720 | 0.465 | 0.445 | **0.457** | **0.445** | 0.486 | 0.447 | **0.457** | **0.438** | 0.486 | 0.456 | **0.465** | **0.447** | 0.455 | 0.441 | **0.451** | **0.440** |
| **ETTm2** 96 | 0.177 | 0.263 | **0.175** | **0.259** | 0.185 | 0.267 | **0.174** | **0.258** | 0.184 | 0.267 | **0.179** | **0.263** | 0.175 | 0.257 | **0.170** | **0.254** |
| 192 | 0.242 | **0.305** | 0.241 | 0.306 | 0.248 | 0.306 | **0.239** | **0.301** | 0.254 | 0.313 | **0.246** | **0.307** | 0.239 | 0.299 | **0.239** | 0.300 |
| 336 | **0.304** | 0.345 | 0.304 | 0.347 | 0.308 | **0.343** | 0.304 | 0.345 | 0.312 | 0.350 | **0.311** | 0.351 | 0.298 | 0.339 | **0.296** | 0.340 |
| 720 | **0.401** | **0.401** | 0.410 | 0.409 | **0.408** | 0.398 | 0.408 | 0.407 | **0.412** | 0.407 | 0.417 | 0.415 | 0.393 | **0.394** | 0.389 | 0.397 |
| **Electricity** 96 | 0.193 | 0.284 | **0.180** | **0.253** | 0.210 | 0.280 | **0.174** | **0.250** | 0.148 | 0.240 | **0.145** | **0.237** | 0.156 | 0.248 | **0.151** | **0.244** |
| 192 | 0.198 | 0.289 | **0.184** | **0.262** | 0.206 | 0.281 | **0.180** | **0.259** | 0.163 | 0.254 | **0.161** | **0.251** | 0.170 | 0.261 | **0.165** | **0.258** |
| 336 | 0.214 | 0.304 | **0.198** | **0.278** | 0.219 | 0.296 | **0.195** | **0.276** | 0.177 | 0.270 | **0.175** | **0.268** | 0.187 | 0.277 | **0.186** | **0.275** |
| 720 | 0.255 | 0.336 | **0.239** | **0.322** | 0.260 | 0.329 | **0.233** | **0.314** | 0.227 | 0.311 | **0.219** | **0.303** | 0.227 | 0.312 | **0.221** | **0.307** |
| **Solar** 96 | 0.233 | 0.287 | **0.210** | **0.263** | 0.336 | 0.351 | **0.231** | **0.270** | 0.205 | 0.238 | **0.196** | **0.235** | 0.198 | 0.261 | **0.195** | **0.248** |
| 192 | 0.266 | 0.307 | **0.241** | **0.286** | 0.376 | 0.371 | **0.272** | **0.292** | 0.237 | 0.262 | **0.235** | **0.257** | 0.241 | 0.274 | **0.238** | 0.275 |
| 336 | 0.291 | 0.317 | **0.261** | **0.297** | 0.415 | 0.384 | **0.301** | **0.313** | 0.251 | 0.275 | **0.248** | **0.275** | 0.253 | 0.274 | **0.235** | **0.273** |
| 720 | 0.286 | 0.316 | **0.263** | **0.300** | 0.413 | 0.374 | **0.297** | **0.310** | 0.250 | 0.276 | **0.246** | **0.274** | 0.231 | 0.271 | **0.225** | **0.271** |
| **Weather** 96 | 0.179 | 0.220 | **0.168** | **0.208** | 0.201 | 0.240 | **0.175** | **0.218** | 0.174 | 0.214 | **0.170** | **0.211** | 0.163 | 0.210 | **0.161** | **0.208** |
| 192 | 0.225 | 0.259 | **0.216** | **0.253** | 0.242 | 0.273 | **0.220** | **0.256** | 0.222 | 0.255 | **0.217** | **0.252** | 0.208 | 0.252 | **0.206** | **0.250** |
| 336 | 0.279 | 0.298 | **0.276** | **0.297** | 0.294 | 0.308 | **0.284** | **0.302** | 0.281 | 0.299 | **0.277** | **0.298** | 0.266 | **0.291** | **0.263** | **0.291** |
| 720 | 0.354 | 0.347 | **0.355** | **0.350** | 0.366 | 0.355 | **0.364** | **0.353** | 0.361 | 0.352 | **0.356** | **0.350** | 0.341 | 0.343 | 0.344 | 0.347 |
| **Traffic** 96 | 0.459 | 0.298 | **0.428** | **0.288** | 0.664 | 0.395 | **0.454** | **0.306** | 0.393 | 0.268 | **0.390** | **0.266** | 0.489 | 0.296 | **0.453** | **0.275** |
| 192 | 0.468 | 0.301 | **0.450** | **0.294** | 0.611 | 0.366 | **0.456** | **0.302** | 0.415 | 0.277 | **0.415** | **0.278** | 0.495 | 0.299 | **0.473** | **0.285** |
| 336 | 0.483 | 0.307 | **0.466** | **0.299** | 0.619 | 0.367 | **0.480** | **0.313** | 0.429 | 0.284 | **0.426** | **0.283** | 0.533 | 0.311 | **0.503** | **0.294** |
| 720 | 0.518 | 0.326 | **0.493** | **0.315** | 0.655 | 0.387 | **0.516** | **0.333** | 0.461 | 0.301 | **0.447** | **0.299** | 0.557 | 0.322 | **0.539** | **0.309** |
| **PEMS03** 12 | 0.085 | 0.196 | **0.074** | **0.183** | 0.145 | 0.258 | **0.081** | **0.191** | 0.069 | 0.174 | **0.067** | **0.172** | **0.063** | **0.168** | 0.064 | 0.170 |
| 24 | 0.135 | 0.249 | **0.105** | **0.217** | 0.267 | 0.352 | **0.126** | **0.238** | 0.098 | 0.209 | **0.092** | **0.202** | **0.084** | **0.195** | 0.084 | 0.196 |
| 36 | 0.180 | 0.287 | **0.134** | **0.246** | 0.416 | 0.449 | **0.178** | **0.285** | 0.130 | 0.243 | **0.119** | **0.231** | **0.099** | 0.213 | **0.099** | **0.212** |
| 48 | 0.231 | 0.326 | **0.165** | **0.276** | 0.574 | 0.541 | **0.229** | **0.328** | 0.164 | 0.275 | **0.149** | **0.260** | **0.110** | 0.224 | **0.110** | **0.223** |
| **PEMS04** 12 | 0.106 | 0.218 | **0.089** | **0.198** | 0.158 | 0.274 | **0.096** | **0.207** | 0.081 | 0.189 | **0.077** | **0.182** | 0.070 | 0.174 | **0.069** | **0.173** |
| 24 | 0.170 | 0.279 | **0.122** | **0.235** | 0.286 | 0.374 | **0.143** | **0.256** | 0.099 | 0.211 | **0.090** | **0.199** | 0.079 | 0.186 | **0.078** | **0.185** |
| 36 | 0.239 | 0.337 | **0.156** | **0.268** | 0.436 | 0.468 | **0.194** | **0.301** | 0.119 | 0.233 | **0.105** | **0.216** | 0.086 | 0.195 | **0.085** | **0.193** |
| 48 | 0.310 | 0.386 | **0.191** | **0.301** | 0.601 | 0.561 | **0.250** | **0.347** | 0.134 | 0.248 | **0.116** | **0.227** | 0.097 | 0.210 | **0.094** | **0.207** |
| **PEMS07** 12 | 0.080 | 0.190 | **0.070** | **0.175** | 0.131 | 0.248 | **0.075** | **0.180** | 0.066 | **0.160** | **0.065** | 0.161 | 0.059 | 0.157 | **0.057** | **0.155** |
| 24 | 0.134 | 0.246 | **0.101** | **0.210** | 0.262 | 0.354 | **0.120** | **0.228** | 0.088 | 0.191 | **0.083** | **0.183** | 0.077 | 0.181 | **0.076** | **0.179** |
| 36 | 0.193 | 0.295 | **0.130** | **0.240** | 0.423 | 0.459 | **0.167** | **0.272** | 0.105 | 0.211 | **0.097** | **0.199** | 0.095 | 0.201 | **0.094** | **0.200** |
| 48 | 0.253 | 0.339 | **0.157** | **0.266** | 0.593 | 0.556 | **0.233** | **0.314** | 0.121 | 0.228 | **0.110** | **0.213** | 0.115 | 0.224 | **0.106** | **0.214** |
| **PEMS08** 12 | 0.097 | 0.208 | **0.088** | **0.194** | 0.150 | 0.266 | **0.094** | **0.201** | 0.081 | 0.185 | **0.079** | **0.184** | 0.079 | 0.184 | **0.078** | **0.182** |
| 24 | 0.153 | 0.259 | **0.126** | **0.233** | 0.274 | 0.365 | **0.148** | **0.254** | 0.123 | 0.227 | **0.115** | **0.221** | 0.113 | 0.224 | **0.111** | **0.218** |
| 36 | 0.217 | 0.313 | **0.168** | **0.272** | 0.425 | 0.462 | **0.205** | **0.305** | 0.167 | 0.264 | **0.153** | **0.256** | 0.142 | 0.250 | **0.141** | **0.246** |
| 48 | 0.278 | 0.356 | **0.207** | **0.304** | 0.598 | 0.559 | **0.272** | **0.357** | 0.217 | 0.305 | **0.194** | **0.288** | 0.184 | 0.292 | **0.180** | **0.283** |

consistently improving the forecasting performance of various backbone models across diverse datasets and settings.

## C Generalizability Investigation

In this section, we conduct comparative experiments on the Energy and Health datasets, which include additional aligned textual descriptions as exogenous information. These experiments aim to assess the generalizability of the proposed PIR framework in handling more complex contextual information. The results, presented in Table 5, show that PIR consistently improves performance in most cases, thereby validating its effectiveness in multimodal forecasting scenarios.

Table 5: Comparison experiments on the Energy and Health datasets. The input series length $L_{in}$ is set to 24, and the target length $L_{out}$ is chosen as {12,24,36,48}. The **bold** values indicate better performance.

| Methods | PatchTST | | + PIR | | SparseTSF | | + PIR | | iTransformer | | + PIR | | TimeMixer | | + PIR | |
|---|---|---|---|---|---|---|---|---|---|---|---|---|---|---|---|---|
| Metric | MSE | MAE | MSE | MAE | MSE | MAE | MSE | MAE | MSE | MAE | MSE | MAE | MSE | MAE | MSE | MAE |
| Energy 12 | 0.194 | 0.334 | **0.130** | **0.255** | 0.140 | 0.273 | **0.128** | **0.255** | 0.122 | 0.245 | **0.117** | **0.242** | 0.149 | 0.286 | **0.135** | **0.263** |
| Energy 24 | 0.299 | 0.418 | **0.248** | **0.364** | 0.252 | 0.375 | **0.241** | **0.362** | 0.242 | 0.365 | **0.232** | **0.355** | 0.226 | 0.348 | **0.222** | **0.346** |
| Energy 36 | 0.382 | 0.475 | **0.337** | **0.432** | 0.335 | 0.436 | **0.321** | **0.419** | 0.327 | 0.426 | **0.309** | **0.414** | 0.319 | 0.422 | **0.314** | **0.420** |
| Energy 48 | 0.463 | 0.527 | **0.425** | **0.496** | 0.418 | 0.493 | **0.416** | **0.493** | 0.420 | 0.493 | **0.406** | **0.485** | 0.408 | 0.487 | **0.393** | **0.477** |
| Health 12 | 12.493 | 1.960 | **9.901** | **1.612** | 11.333 | 1.802 | **10.315** | **1.641** | 8.523 | 1.469 | **8.258** | **1.461** | 8.947 | 1.493 | **8.642** | **1.489** |
| Health 24 | 15.252 | 2.204 | **13.486** | **1.924** | 14.115 | 2.077 | **13.837** | **1.975** | 12.159 | 1.801 | **11.698** | **1.748** | 12.010 | 1.789 | **11.790** | **1.765** |
| Health 36 | 13.804 | 2.130 | **12.032** | **1.893** | **12.856** | 2.052 | 12.876 | **1.969** | 11.662 | 1.854 | **11.093** | **1.793** | 11.385 | 1.801 | **11.029** | **1.792** |
| Health 48 | 12.515 | 2.062 | **12.294** | **1.965** | 13.167 | 2.015 | **11.784** | **2.001** | 10.910 | 1.817 | **10.332** | **1.781** | 11.227 | 1.879 | **10.328** | **1.784** |

# D  Ablation Study

In this section, we investigate the impact of the auxiliary constraint and structural designs on the overall performance of the PIR framework through ablation studies. The backbone models selected for this analysis are PatchTST and iTransformer, representing the channel-independent and channel-dependent categories, respectively.

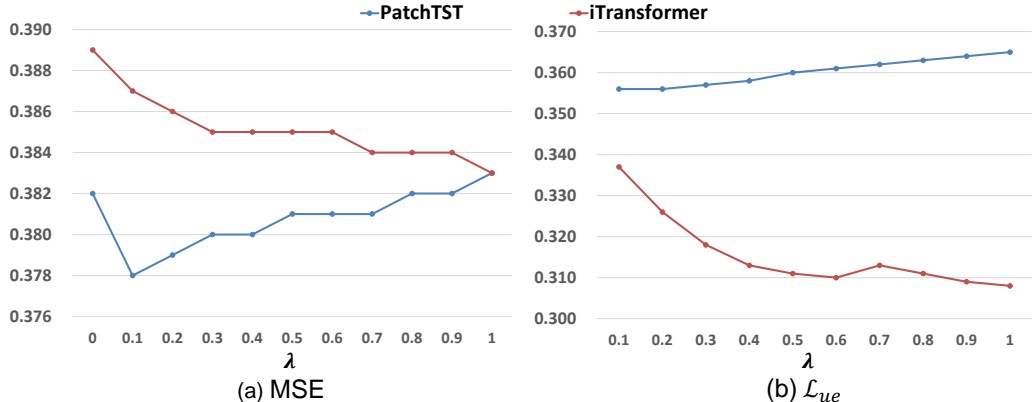

Figure 5: MSE and the uncertainty estimation error, $\mathcal{L}_{ue}$, of PIR-enhanced PatchTST and iTransformer on the ETTm1 dataset with different $\lambda$s.

Table 6: MAE comparison between different variants of the PIR framework. The **Bold** values indicate the best performance.

| Backbone | | PatchTST | | | | iTransformer | | |
|---|---|---|---|---|---|---|---|---|
| Variants | w/o PIR | w/o Local | w/o Global | w/ PIR | w/o PIR | w/o Local | w/o Global | w/ PIR |
| Electricity 96 | 0.284 | 0.257 | 0.270 | **0.253** | 0.240 | 0.238 | 0.241 | **0.237** |
| Electricity 192 | 0.289 | 0.266 | 0.277 | **0.262** | 0.254 | 0.253 | 0.256 | **0.251** |
| Electricity 336 | 0.304 | 0.282 | 0.292 | **0.278** | 0.270 | **0.268** | 0.272 | **0.268** |
| Electricity 720 | 0.336 | 0.326 | 0.325 | **0.322** | 0.311 | 0.305 | 0.304 | **0.303** |
| PEMS03 12 | 0.196 | 0.201 | 0.192 | **0.183** | 0.174 | 0.175 | 0.174 | **0.172** |
| PEMS03 24 | 0.249 | 0.243 | 0.228 | **0.217** | 0.209 | 0.204 | 0.207 | **0.202** |
| PEMS03 36 | 0.287 | 0.281 | 0.262 | **0.246** | 0.243 | 0.234 | 0.239 | **0.231** |
| PEMS03 48 | 0.326 | 0.314 | 0.298 | **0.276** | 0.275 | 0.262 | 0.272 | **0.260** |

To evaluate the impact of the auxiliary constraint, we compare forecasting performance across various values of the weight hyperparameter $\lambda$, ranging from 0 to 1. Additionally, we report the corresponding uncertainty estimation error $\mathcal{L}_{ue}$, as defined in Equation 1, in Figure 5. The results suggest that the auxiliary constraint enables the PIR framework to more accurately estimate the

uncertainty of intermediate forecasts by predicting their associated errors, thereby facilitating the revision weight learning, leading to better accuracy. Furthermore, the R-squared scores between MSE and $\mathcal{L}_{ue}$ are **0.9067** and **0.7500** when PatchTST and iTransformer are used as the backbone models, respectively. These high correlations strongly support the feasibility of our approach, validating the use of forecasting error as a proxy for uncertainty in time series forecasting tasks.

On the other hand, we present the MAE comparison between different variants of the PIR framework on the Electricity and PEMS03 datasets in Table 6, providing an intuitive understanding of how the local and global revision components contribute to the performance. It can be inferred from the results that both components contribute to better performance compared to the baseline model under most cases, and the combination of them consistently leads to the best accuracy, validating our proposal that utilizing contextual information to revise forecasting results from both local and global perspectives can well alleviate the impact of the instance-level variance.

To further examine where the performance gains of the PIR framework originate, we conduct an ablation study comparing PIR with two variants: (1) a deepened iTransformer, which increases model depth to match the additional layers introduced by PIR's local revision component, and (2) PIR trained from scratch, which removes the requirement of a pretrained forecasting backbone. The results on the Electricity and Weather datasets are presented in Table 7. The results show that PIR consistently outperforms both variants. Benefiting from both global and local revision components, PIR effectively utilizes retrieved similar historical series as well as valuable contextual insights from covariates and exogenous variables. This allows PIR to outperform simply enlarging model capacity. Furthermore, training PIR from scratch leads to degraded performance, possibly because the randomly initialized backbone produces unstable forecasts in early training stages, which negatively affects the optimization of the failure identification module and weakens its ability to estimate forecasting error.

Table 7: Investigation into the sources of performance improvement. The **Bold** values indicate the best performance.

| Variants Metric | | iTransformer | | w/ PIR | | w/ Deepen | | w/ PIR scratch | |
|---|---|---|---|---|---|---|---|---|---|
| | | MSE | MAE | MSE | MAE | MSE | MAE | MSE | MAE |
| Electricity | 96 | 0.148 | 0.240 | **0.145** | **0.237** | 0.147 | 0.239 | 0.164 | 0.256 |
| | 192 | 0.163 | 0.254 | **0.161** | **0.251** | 0.163 | 0.255 | 0.178 | 0.270 |
| | 336 | 0.177 | 0.270 | **0.175** | **0.268** | 0.177 | 0.271 | 0.196 | 0.284 |
| | 720 | 0.227 | 0.311 | 0.219 | **0.303** | **0.217** | 0.304 | 0.256 | 0.334 |
| Weather | 96 | 0.174 | 0.214 | **0.170** | **0.211** | 0.177 | 0.218 | 0.217 | 0.249 |
| | 192 | 0.222 | 0.255 | **0.217** | **0.252** | 0.224 | 0.257 | 0.252 | 0.270 |
| | 336 | 0.281 | 0.299 | **0.277** | 0.298 | 0.279 | **0.296** | 0.300 | 0.315 |
| | 720 | 0.361 | 0.352 | **0.356** | **0.350** | 0.358 | **0.350** | 0.393 | 0.371 |

# E  Forecasting showcases

In this section, we provide intuitive forecasting showcases to illustrate the impact of the revision process on forecasting performance, and the effect of global revision in cases with and without similar historical instances.

In Figure 6, we examine the impact of the revision process on forecasting results using PatchTST as the backbone model across three datasets of varying scales. For the local revision component, it struggles to capture scale changes using only cross-channel dependencies and exogenous information on the ETTh1 dataset, but successfully corrects trend deviations to better align with the ground truth on the Solar dataset. Meanwhile, the global revision component improves future scale predictions by retrieving similar historical series on the ETTh1 dataset but has minimal impact on the Solar dataset. Furthermore, both local and global revisions contribute to improved forecasting accuracy on the PEMS04 dataset, with their combination (i.e., the PIR framework) achieving superior performance. These findings indicate that local and global revisions each have their strengths and limitations, underscoring the importance of constructing a unified framework that effectively integrates both approaches.

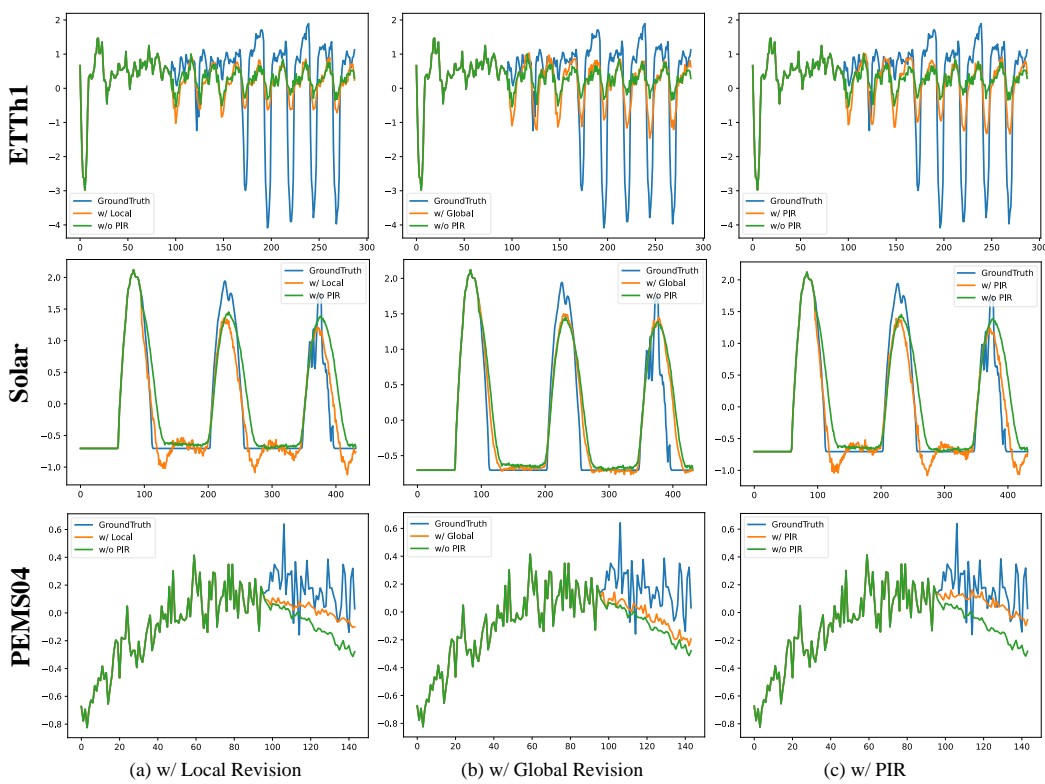

Figure 6: Illustration of forecasting showcases of different variants of PIR using PatchTST as the backbone model.

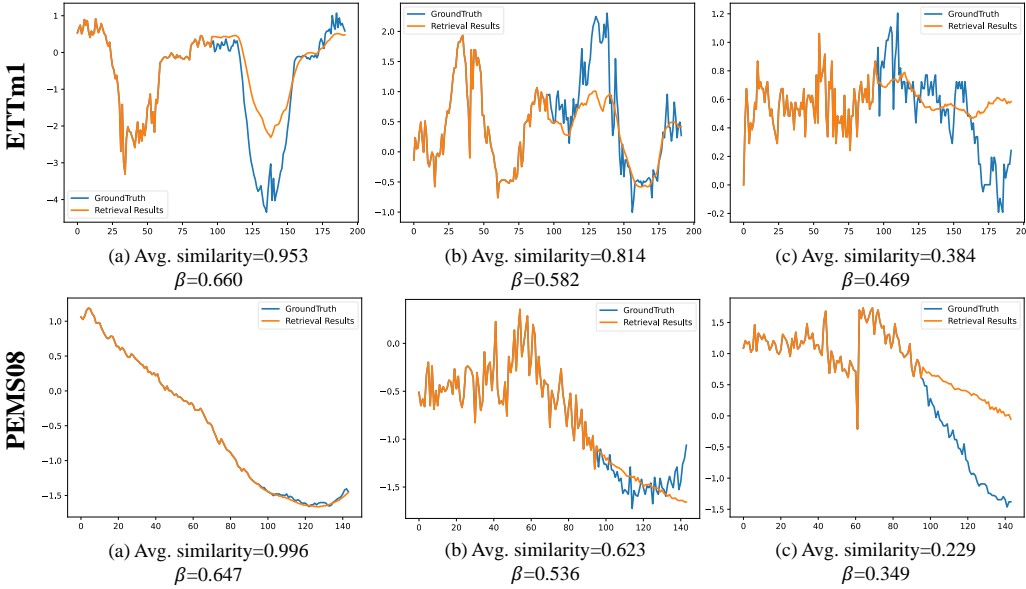

Figure 7: Illustration of forecasting showcases on various forecasting instances with and without similar historical time series.

Additionally, we evaluate the performance of global revision with and without similar historical instances. In Figure 7, we present the retrieval results ($y_{global}$, as defined in Equation 4 of our paper) as the forecasting outputs, along with the corresponding average top-k similarities and learned weight $\beta$. The results indicate that retrieval similarities vary significantly across instances. For instances with highly similar historical series, the retrieval process effectively captures future evolutions, whereas for those without close historical analogs, the retrieval results only approximate future trends and may even be inaccurate. This observation aligns with our expectations, as the retrieval mechanism is designed to leverage similar past series as references for forecasting. Furthermore, the PIR framework adaptively assigns higher weights to instances with stronger retrieval similarities, dynamically adjusting the influence of retrieval results on the final forecast.

## F   Limitations

Though PIR demonstrates promising performance on benchmark datasets, there are still several limitations within this framework. Firstly, the instance-level variances exist in both input series and target series, while the framework currently addresses only the former challenge. Identifying and addressing instance-level variances in the target series caused by data quality issues—such as noise, outliers, and missing data—holds significant potential for improving both the training and evaluation processes. Furthermore, relatively simple network architectures are utilized in the failure identification and local revision components. Exploring ways to integrate advanced inductive biases into these components to enhance their performance remains an important direction for future research.

