# OpenReview forum: "Improving Time Series Forecasting via Instance-aware Post-hoc Revision"
_NeurIPS.cc/2025/Conference — NeurIPS 2025 poster_

### Official Review · Reviewer_6UsG · 2025-06-29

**Clarity:** 3
**Significance:** 3
**Originality:** 3
**Rating:** 5
**Confidence:** 5

**Summary:**

This paper proposes PIR, a post-hoc refinement method for time series forecasting that improves accuracy through instance-level revision. PIR first identifies potentially erroneous predictions using an auxiliary regressor trained to estimate MAE. Then it revises the initial prediction in two ways: a local revision uses a Transformer to refine the initial forecast based on its own output, and a global revision retrieves relevant instances from the training data to guide correction.

**Questions:**

1. I want to know what happens if the loss terms are included in the original training process. (W1)

2. Please clarify that uncertainty estimator's prediction really align with the actual error, and its projections do not deteriorate the measured uncertainty. (W3)

3. Please compare with linear baselines (W4)

4. And other miscellaneous paper works (W2, W5)

**Ethical Concerns:**

["NO or VERY MINOR ethics concerns only"]

**Final Justification:**

After reviewing the authors’ rebuttal and additional experiments, I am increasing my score. The authors addressed key concerns with detailed and well-supported responses:

1) Effectiveness of PIR: Ablation studies show that PIR outperforms simple model scaling and training from scratch, validating its design and two-stage training.

2) Uncertainty estimation: The authors demonstrated strong alignment between estimated and actual errors (e.g., estimation error 0.1752 vs. MSE 0.498), and explained the learned projection behavior clearly.

3) Generality: PIR improves performance across additional models (DLinear, RAFT), confirming its generalizability beyond transformer-based backbones.

Overall, the concerns have been sufficiently addressed, and the contribution is now clearer and stronger.

**Limitations:**

Yes

**Quality:**

3

**Strengths And Weaknesses:**

Strengths

1. The high-level idea is promising, and the methodology is relevant and well-aligned with the goal.

2. The paper is clearly written and easy to follow.

Weaknesses

1. It is necessary to verify whether the observed performance gains are simply due to increased model capacity or training time, particularly in the case of the local revision module. What happens if the loss terms are incorporated into the original training process, instead of post-processing?

2. For global revision, the idea of retrieving historical patches and computing a weighted sum is closely related to RAFT. A methodological comparison with RAFT would strengthen the paper.

3. The paper lacks a detailed evaluation of the uncertainty estimator's performance. In addition, unless explicit constraints are applied during projection, the direction of the projection vector may change. This could lead to situations where increasing $\delta$ results in decreased $\alpha$ or $\beta$. If no constraint is enforced in the implementation, experimental validation of this behavior is needed.

4. While the use of strong Transformer-based baselines is appreciated, linear baselines should also be included. DLinear has shown that linear models can perform competitively with Transformers, and RAFT extends this by incorporating retrieval. Including these would provide a more balanced and informative comparison. (RAFT is a very recent work, so while its inclusion would be beneficial, it is also reasonable if it is omitted.)

5. The current qualitative analysis (Figure 3) is useful, but the showcase in Section F (Figure 6) is even more compelling. Including this example in the main paper would make the contribution more impactful.

[1] Retrieval augmented time series forecasting (ICML 2025)
[2] Are transformers effective for time series forecasting? (AAAI 2023)

---

> ### Author Rebuttal · Authors · 2025-07-29
>
> Thank you for your acknowledgement and valuable feedbacks on our work, we would like to address your concerns as follows.
>
> - [Q1] We provide additional comparison experiments as shown below. Here, “**Increase model capacity**” refers to an iTransformer model with comparable parameter count to the PIR-enhanced version, and “**+PIR (scratch)**” denotes the variant where PIR’s loss terms are incorporated from the start without pretraining the forecasting backbone:
>
>   |             |      | iTransformer |       | +PIR  |       | Increase model capacity |       | +PIR(scratch) |       |
>   | ----------- | ---- | ------------ | ----- | ----- | ----- | ----------------------- | ----- | ------------- | ----- |
>   |             |      | MSE          | MAE   | MSE   | MAE   | MSE                     | MAE   | MSE           | MAE   |
>   | Electricity | 96   | 0.148        | 0.240 | 0.145 | 0.237 | 0.147                   | 0.239 | 0.164         | 0.256 |
>   |             | 192  | 0.163        | 0.254 | 0.161 | 0.251 | 0.163                   | 0.255 | 0.178         | 0.270 |
>   |             | 336  | 0.177        | 0.270 | 0.175 | 0.268 | 0.177                   | 0.271 | 0.196         | 0.284 |
>   |             | 720  | 0.227        | 0.311 | 0.219 | 0.303 | 0.217                   | 0.304 | 0.256         | 0.334 |
>   | Weather     | 96   | 0.174        | 0.214 | 0.170 | 0.211 | 0.177                   | 0.218 | 0.217         | 0.249 |
>   |             | 192  | 0.222        | 0.255 | 0.217 | 0.252 | 0.224                   | 0.257 | 0.252         | 0.279 |
>   |             | 336  | 0.281        | 0.299 | 0.277 | 0.298 | 0.279                   | 0.296 | 0.300         | 0.315 |
>   |             | 720  | 0.361        | 0.352 | 0.356 | 0.350 | 0.358                   | 0.350 | 0.393         | 0.371 |
>
>   Benefiting from both global and local revision components, PIR effectively utilizes retrieved similar historical series as well as valuable contextual insights from covariates and exogenous variables. This allows PIR to outperform simply enlarging model capacity. Furthermore, training PIR from scratch leads to degraded performance, possibly because the randomly initialized backbone produces unstable forecasts in early training stages, which negatively affects the optimization of the failure identification module and weakens its ability to estimate forecasting error.
>
> - [Q2]
>
>   - We illustrate the alignment between the uncertainty estimator’s prediction and actual forecasting errors in Fig. 3 of our paper. Here we mistakenly place the suboptimal results on the Weather dataset (compared to the results on the Solar dataset, and we will correct the mistake in the revised paper), yet the estimated and actual error curves exhibit consistent patterns in terms of peaks and troughs, which validates PIR’s capability to assess the quality of individual forecasting results. To be more specific, the estimation error $L_{ue}$, defined in Eq1 in our paper, on the test set of linear estimator is only **0.1752** when $L_{in}=96$,$L_{out}=720$ on the ETTh1 dataset using PatchTST as the backbone model, which is much smaller than the MSE (0.498) of backbone model, demonstrating that the uncertainty estimator's prediction really align with the actual error.
>   - Besides, your concern on the projections is of great value. We do not include explicit constraints on these projections, while the learned results indeed meet the expectations. For example, the weights of $\alpha$'s projectin learned on the ETTh1 dataset of 4 target lengths are [0.9579, 0.9620, 0.9699, 0.9776] respectively, ensuring a positive correlation between $\alpha$ and $\delta$. Since the global revision process considers both estimated uncertainty and retrieved similarity, we do not enforce a simple positive correlation between $\beta$ and $\delta$. Instead, we employ an MLP to learn their relationship. As shown in the case study in Fig. 7, the model tends to assign higher $\beta$ values to samples with higher average similarity, which is consistent with our expectations.
>
> - [Q3&Q4]
>
>   - Our experiments have already included a linear-based forecasting model, SparseTSF, for comparison. Here we also present additional experiments on the traditional DLinear model and the suggested RAFT model as follows. Note that we fix $L_{in}=96$ therefore the results may have differences.
>
>     |       |      | DLinear |       | +PIR  |       | RAFT  |       |
>     | ----- | ---- | ------- | ----- | ----- | ----- | ----- | ----- |
>     |       |      | MSE     | MAE   | MSE   | MAE   | MSE   | MAE   |
>     | ETTm1 | 96   | 0.318   | 0.376 | 0.284 | 0.336 | 0.288 | 0.340 |
>     |       | 192  | 0.427   | 0.441 | 0.366 | 0.389 | 0.384 | 0.401 |
>     |       | 336  | 0.466   | 0.468 | 0.412 | 0.424 | 0.431 | 0.445 |
>     |       | 720  | 0.748   | 0.623 | 0.418 | 0.438 | 0.434 | 0.459 |
>     | Solar | 96   | 0.284   | 0.373 | 0.235 | 0.276 | 0.226 | 0.287 |
>     |       | 192  | 0.315   | 0.393 | 0.277 | 0.299 | 0.251 | 0.311 |
>     |       | 336  | 0.349   | 0.411 | 0.304 | 0.320 | 0.273 | 0.339 |
>     |       | 720  | 0.354   | 0.410 | 0.300 | 0.322 | 0.266 | 0.343 |
>
>     It can be inferred that our PIR demonstrates generalizability on the DLinear model, yielding significant performance improvements. Besides, even compared with the most recent approach, our framework can achieve competitive results.
>
>   - Thank you for pointing out the related work. We will include a detailed discussion of the RAFT method in the revised paper. While RAFT also adopts a retrieval-based strategy—enhancing diversity through multi-period retrieval—it ultimately builds a linear forecasting model. In contrast, our method is not designed to propose a new forecasting architecture. Instead, it is motivated by the observation of instance-level variance and aims to build a general framework that corrects potential prediction failures of backbone models from a post-processing perspective. To this end, we introduce two components: a global retrieval component and a local covariate modeling component. The global retrieval component is conceptually similar to the retrieval module in RAFT. Moreover, we share the same insight as RAFT by employing different strides in the retrieval process to balance between pattern diversity and computational efficiency.
>
> We hope our responses adequately address your concerns, and we sincerely appreciate the opportunity to clarify and improve our work. We will include the suggested experiments, discussions and in-depth cases in our revised paper.

---

> > ### Comment · Reviewer_6UsG · 2025-08-03
> >
> > Thank you for the detailed and constructive rebuttal. Your clarifications and additional results addressed my concerns, and I have raised my score accordingly.

---

### Official Review · Reviewer_PruS · 2025-07-02

**Clarity:** 2
**Significance:** 2
**Originality:** 2
**Rating:** 5
**Confidence:** 5

**Summary:**

The paper proposes a model-agnostic and extensible framework for improving time series forecasting. It identifies prediction failure cases through uncertainty estimation and addresses them from both local and global perspectives. Experimental results demonstrate the framework's effectiveness.

**Questions:**

- What is the specific retrieval process during model training? Does
the model start with a complete training set as the retrieval database, or does it use a slow-start mechanism to gradually accumulate data? Is the training process sequential or random? How do these differences affect the final results?
- The framework shows significant performance variations across different models and datasets. Why are there differences between datasets and models? Does the model architecture (e.g., transformer- based or linear-based) have an impact?
- Referring to the strengths and weaknesses section, could you provide more thorough analysis on the failure identification part? For example, more comprehensive theoretical support, how the lag effect of covariates works and to what extent it can identify different failure types, and case examples with more detailed analysis? Please note that since I consider this part critical to the paper, this response is crucial to me.

**Ethical Concerns:**

["NO or VERY MINOR ethics concerns only"]

**Final Justification:**

According to the authors’ rebuttal and additional experiments, I increase my score. Thanks for the additional explanation.

**Limitations:**

yes.

**Paper Formatting Concerns:**

no major formatting issues

**Quality:**

2

**Strengths And Weaknesses:**

strengths:
- Easy to extend to existed methods.
- The experimental results show effectiveness.
Weaknesses:
-  Limited improvement on strong baselines like timemixer
- Lack of some training and experimental details and related analysis,
   e.g., retrieval process (see questions for details)
- Lack of theoretical or case analysis: Given the forecasting scenario,
the key to whether the entire framework works lies in whether prediction failure cases can be accurately identified in advance. Meanwhile, the revision stage methods are existing approaches in this field [1], so I pay more attention to the failure identification methods and related analysis. Unfortunately, the authors also use a simple method (mlp) here with insufficient theoretical analysis or case studies. Specifically, the paper classifies prediction failure causes but lacks detailed analysis of each category; the key factor proposed for uncertainty estimation is the lag of covariates, but this part also lacks in-depth discussion; finally, the paper only uses sample pairs and covariates to predict uncertainty, but its sufficiency remains questionable. For example, in distribution shift scenarios caused by weekdays and holidays, historical sequences alone may not be sufficient to identify prediction failures. Would incorporating timestamp changes or other external information as inputs for uncertainty prediction be more reasonable?
[1] RATSF: Empowering Customer Service Volume Management through Retrieval-Augmented Time-Series Forecasting

---

> ### Author Rebuttal · Authors · 2025-07-29
>
> Thank you for your acknowledgement and valuable feedbacks on our work, we would like to address your concerns as follows.
>
> - [Q1&W2] To fully utilize the available time series data for better capturing the underlying data patterns, we use the entire training set as the retrieval database for training with a random process(termed as offline training). To avoid data leakage, we mask any portion of the historical data that overlaps with the query. Here we also investigate the impact of only retrieving on the previous time series data for a given query(termed as online training) as follows, and the results show that the online training strategy achieves comparable performance with the offline one:
>
>   |        |      | PatchTST |       | +PIR(offline) |       | +PIR(online) |       |
>   | ------ | ---- | -------- | ----- | ------------- | ----- | ------------ | ----- |
>   |        |      | MSE      | MAE   | MSE           | MAE   | MSE          | MAE   |
>   | ETTh2  | 96   | 0.299    | 0.347 | 0.291         | 0.340 | 0.292        | 0.343 |
>   |        | 192  | 0.384    | 0.398 | 0.371         | 0.391 | 0.371        | 0.394 |
>   |        | 336  | 0.424    | 0.432 | 0.418         | 0.429 | 0.420        | 0.431 |
>   |        | 720  | 0.427    | 0.444 | 0.421         | 0.441 | 0.425        | 0.443 |
>   | PEMS03 | 12   | 0.085    | 0.196 | 0.074         | 0.183 | 0.075        | 0.181 |
>   |        | 24   | 0.135    | 0.249 | 0.105         | 0.217 | 0.103        | 0.216 |
>   |        | 36   | 0.180    | 0.287 | 0.134         | 0.246 | 0.135        | 0.246 |
>   |        | 48   | 0.231    | 0.326 | 0.165         | 0.276 | 0.166        | 0.276 |
>
> - [Q2&W1]
>
>   - The backbone architecture indeed plays a significant role in forecasting accuracy, with simpler models (e.g., linear-based) generally underperforming more advanced ones. As a plug-in method, the strength of our PIR framework lies in its ability to improve forecasting accuracy across a variety of backbones while also mitigating performance variance caused by architectural differences. This explains why our method yields substantial improvements on certain models, while the gains on others are relatively modest.
>   - As for the performance differences across datasets, we attribute this to inherent characteristics of each dataset. While PIR addresses instance-level variations often overlooked by prior methods, it cannot fully account for uncertainties stemming from dataset construction—such as imputation or smoothing— which may cap the achievable performance. These data-specific factors may introduce an upper bound on the achievable accuracy, which may in turn limit the performance improvement attainable by our method.
>
> - [Q3&W3]
>
>   - Thank you for pointing out the related work. We will include a detailed discussion of the RATSF method in the revised paper. While both RATSF and our PIR framework leverage retrieval-based designs to enhance forecasting, the key difference lies in our design goals: PIR is a model-agnostic, post-processing framework that revises initial forecasting results from any backbone. To achieve this, we introduce both global and local revision components—via weighted aggregation of retrieved series and cross-channel attention over covariates and exogenous variables—making PIR fundamentally different from RATSF.
>   - We agree with your insight that the failure identification component is critical in our work. Although we have identified several potential reasons that may lead to prediction failure, there are many other potential and interrelated factors, and there is a lack of ground truth and metrics to identify and disentangle the specific reasons for each forecasting failure instance. **Therefore, we do not aim to explicitly locate these patterns to identify forecasting failures**. Instead, we define failure instances as those with high forecasting errors, which are then prioritized for revision. This approach is more flexible and can potentially accommodate a broader range of failure modes. Besides, we appreciate your suggestion to incorporate external information into the failure identification module. However, such information is often only relevant for a limited subset of failure cases, and incorporating them may introduce noise. Therefore, our current design relies solely on the original time series, immediate forecasts, and channel indicators to maintain robustness. We plan to explore more sophisticated strategies in future work.
>   - To further validate the effectiveness and rationality of our proposal of using MLP for failure identification, we present the comparison experiment of transformer-based uncertainty estimator as follows. The results demonstrate that the simple MLP layers can well handle the task of forecasting error estimation. Moreover, the estimation error $L_{ue}$, defined in Eq1 in our paper, on the test set of linear estimator is only **0.1752** when $L_{out}=720$, which is much smaller than the MSE of backbone model (0.498), meaning that the estimator can well assess the quality of individual forecasting results. Similar conclusions can be drawn from the case illustration in Fig3 in our paper, where the estimated and actual error curves exhibit consistent patterns in terms of peaks and troughs.
>
>   | ETTh1/MSE | $L_{in}$=96 |                   |                |
>   | --------- | ----------- | ----------------- | -------------- |
>   |           | PatchTST    | +Linear Estimator | +TRM Estimator |
>   | 96        | 0.410       | 0.375             | 0.374          |
>   | 192       | 0.458       | 0.422             | 0.425          |
>   | 336       | 0.498       | 0.467             | 0.468          |
>   | 720       | 0.498       | 0.484             | 0.490          |
>
>   - Finally, we propose that the lead-lag effects may help to produce accurate results and therefore **the lags are not the key for uncertainty estimation but for the revision process**. Instead of explicitly modeling lead-lag relationships, our local revision component leverages a channel-wise attention mechanism to capture diverse and informative patterns across covariates and exogenous variables. In particular, we incorporate available timestamps to guide the revision process in scenarios with potential distribution shifts—such as those induced by weekdays and holidays—as discussed in Line 234 of the main paper. Further explorations on contextual information modeling are provided in Section 3 of the appendix.
>
> We hope our responses adequately address your concerns, and we sincerely appreciate the opportunity to clarify and improve our work. We will include the suggested experiments, discussions and in-depth cases in our revised paper.

---

### Official Review · Reviewer_X6vo · 2025-07-02

**Clarity:** 3
**Significance:** 3
**Originality:** 3
**Rating:** 4
**Confidence:** 4

**Summary:**

This paper proposes a model-agnostic framework named PIR, which aims to enhance time series forecasting performance through post-forecasting failure identification and revision. PIR identifies biased forecasting instances by estimating their accuracy and revises the forecasts using contextual information, including covariates and historical time series, from both local and global perspectives. Extensive experiments on real-world datasets demonstrate that PIR effectively mitigates instance-level errors and significantly improves forecasting reliability.

**Questions:**

Seen in the weaknesses.

**Ethical Concerns:**

["NO or VERY MINOR ethics concerns only"]

**Final Justification:**

The answers have solved my concern

**Limitations:**

Seen in the weaknesses.

**Quality:**

3

**Strengths And Weaknesses:**

strong
(1) The PIR framework approaches the problem from a novel post processing perspective, focusing on identifying and revising forecasting failures, offering a unique solution in the field of time series forecasting. As a model agnostic plugin, PIR can be seamlessly integrated into any forecasting model, showcasing broad applicability and enhancing its versatility across different architectures.
(2) Experimental results on multiple real world datasets and mainstream forecasting models indicate that PIR can significantly reduce forecasting errors, thereby improving the accuracy and reliability of predictions. In addition, this work provides the first in depth analysis of the performance of existing forecasting methods at the instance level, revealing the impact of instance level variations on forecasting failures and offering new insights for future research. And by estimating forecasting errors to quantify uncertainty and aligning it with actual prediction errors, PIR accurately identifies potential forecasting failure cases.

weak:
(1) Uses a 2-layer MLP but no exploration of advanced architectures (e.g., Transformers).
(2) Brute-force similarity search and no approximate methods (e.g., LSH) for large-scale data.
(3) α/β derived via Sigmoid without theoretical guarantees and fixed λ=1 may be suboptimal (Fig. 5).
(4) Fixed input length (L_in=96) and untested on very long sequences (e.g., L_in>720). Also there is no variance across runs (e.g., random seeds).
(5) No theoretical time complexity analysis (e.g., local revision’s Transformer cost, O(N) retrieval comparisons). The experiments mention "single GPU" but omit overhead comparison (e.g., retrieval latency, memory usage vs. base models).

---

> ### Author Rebuttal · Authors · 2025-07-29
>
> Thank you for your acknowledgement and valuable feedbacks on our work, we would like to address your concerns as follows.
>
> - [W1&W4] Thank you for raising these points. We would like to clarify a few aspects first. In our submitted manuscript, we followed common practice by employing a simple two-layer MLP to model complex dynamics for forecasting failure identification. The input length $L_{in}$ was set to 96 primarily to align with the experimental settings of prior works. Additionally, our main experiments were conducted over three runs, and we report only the average performance. This is due to the stability of recent forecasting models, where the variance across runs is typically negligible.
>
>   We also appreciate your suggestion to explore the generalizability of PIR under different uncertainty estimators and input settings. Below, we provide additional MSE results using PatchTST as the backbone on the ETTh1 dataset:
>
>   | ETTh1/MSE | $L_{in}$=96 |                   |                | $L_{in}$=720 |                   |                |
>   | --------- | ----------- | ----------------- | -------------- | ------------ | ----------------- | -------------- |
>   |           | PatchTST    | +Linear Estimator | +TRM Estimator | PatchTST     | +Linear Estimator | +TRM Estimator |
>   | 96        | 0.410       | 0.375             | 0.374          | 0.384        | 0.378             | 0.380          |
>   | 192       | 0.458       | 0.422             | 0.425          | 0.428        | 0.419             | 0.416          |
>   | 336       | 0.498       | 0.467             | 0.468          | 0.457        | 0.443             | 0.449          |
>   | 720       | 0.498       | 0.484             | 0.490          | 0.505        | 0.493             | 0.512          |
>
>   - First, increasing the look-back window generally improves the baseline model’s accuracy. When enhanced by our PIR framework, performance is further improved, demonstrating its generalizability across different input lengths.
>   - Second, we replaced the default linear estimator with a one-layer transformer block (with roughly tuned hyperparameters) for forecasting failure identification. The results show comparable performance across most settings. However, possibly due to overfitting induced by the transformer's increased capacity, it sometimes underperforms the simpler linear estimator. Specifically, for the case where $L_{in}=96$ and $L_{out}=720$, the estimation error $L_{ue}$ (defined in Eq.1 of our paper) on the test set is 0.1752 for the linear estimator, compared to 0.1770 for the transformer. The results demonstrate that the simple MLP layers can well handle the task of forecasting error estimation, validating the feasibility of our current implementation.
>
> - [W2&W5] We appreciate your suggestions of comparing with approximate methods for retrieving and including theoretical time complexity analysis for our framework.  Theoretically, the series-wise cosine similarity function used in the retrieval process has a complexity of $O(NML_{in})$, where $M$ is the total number of historical series for one channel and $N$ is the number of channels. Meanwhile, the local revision process has a complexity of $O(N^2)$ due to the channel-wise attention mechanism.
>   We further include inference time evaluations on both small-scale (ETTh1) and large-scale (Traffic) datasets using brute-force cosine similarity and LSH:
>
>   |                   | ETTh1(Cosine) | ETTh1(LSH) | Traffic(Cosine) | Traffic(LSH) |
>   | ----------------- | ------------- | ---------- | --------------- | ------------ |
>   | Backbone          | 0.164         | 0.164      | 0.424           | 0.424        |
>   | $\Delta$retrieval | 0.024         | 0.415      | 0.079           | 87.957       |
>   | $\Delta$revision  | 0.096         | 0.096      | 0.275           | 0.275        |
>   | $\Delta$MSE       | -0.014        | -0.009     | -0.025          | -0.025       |
>
>   - Thanks to the GPU-parallelizable nature of cosine similarity, the retrieval process incurs negligible additional inference time on both datasets. Moreover, as analyzed before, for even larger datasets, we can introduce sampling strategies (such as stride) and embedding methods to reduce the total number of historical series and dimensionality to further limite the search space, therefore to reduce the time and memory overhead.
>   - Besides, we also implement the LSH method for similarity computing. Note that the current implementation stores $N$ databases and conduct similarity search using CPUs, therefore it requires much more additional time to retrieval time series data. Moreover, the results show that using the approximate method may only achieve just comparable performance with the cosine similarity, further demonstrating the feasibility of our current implementation.
>
> - [W3]
>
>   - As discussed in our response to [W1 & W4], the linear estimator effectively captures the immediate forecasting error $\delta$, and the $\alpha, \beta$ derived via simple linear layers with sigmoid function ensures a strong correlation between these parameters. This enables effective guidance for the subsequent revision step. To be specific, the weights of $\alpha$'s projectin learned on the ETTh1 dataset of 4 target lengths are [0.9579, 0.9620, 0.9699, 0.9776] respectively, ensuring a positive correlation between $\alpha$ and $\delta$. Besides, since the global revision process should take both the uncertainty and retreived similarity into account, we introduce a MLP layer to model the relationship. The case study in Fig7 illustrates that the PIR framework tends to allocate higher $\beta$ for higher averaged similarities, which makes sense. Consequently, our approach effectively utilizes $\delta$ as a guiding signal for the latter revision process.
>   - We agree that using a fixed $\lambda=1$ may not be optimal. We plan to investigate adaptive strategies for determining $\lambda$ in future work.
>
> We hope our responses adequately address your concerns, and we sincerely appreciate the opportunity to clarify and improve our work. We will include the suggested experiments, discussions and in-depth cases in our revised paper.

---

> > ### Author Response · Authors · 2025-08-08
> >
> > Dear Reviewer X6vo,
> >
> > I hope this message finds you well. Since the discussion period is approaching its end, I wanted to ensure we have addressed all your concerns satisfactorily. If there are any additional points or feedback you'd like to consider, please let us know, which is invaluable to us.
> >
> > Thank you for your time and effort in reviewing our paper.

---

> > > ### Comment · Area_Chair_NaSa · 2025-08-08
> > > **Action Required: Please Discuss Author Rebuttal by 8/8 (AoE)**
> > >
> > > Dear Reviewer,
> > >
> > > Please review the authors' rebuttal and engage in the follow-up discussion. The deadline for this stage is August 8th (AoE). We look forward to your thoughts on their response.
> > >
> > > Thanks,

---

> > ### Comment · Reviewer_X6vo · 2025-08-09
> >
> > Thanks for the reply. The rebuttal has solved my concern. I will adjust my score.

---

### Note · Authors · 2025-08-13

Dear SAC, AC, and the reviewers,

We sincerely thank you for your invaluable expertise and dedication throughout the review process. We hope the following final remarks could clarify our paper’s contributions and summarize the rebuttals, therefore serving as a constructive reference for decision-making.

**Contributions**

In this work, we identify **instance-level variations** in time series forecasting tasks that can lead to suboptimal predictions for specific instances. To address this issue, we propose PIR, a novel, model-agnostic, post-hoc revision framework that **identifies potential forecasting failures and corrects them using contextual information from both local and global perspectives**.

We are grateful to the reviewers for **acknowledging the merits of our work**, including:

- The high-level idea and the proposed solution are promising. (X6vo, 6UsG)
- The PIR framework is flexible and can be easily extended to existing methods. (X6vo, PruS)
- Extensive experiments demonstrate the effectiveness of the PIR framework. (X6vo, PruS)
- The paper is clearly written and easy to follow. (6UsG)

**Summary of the rebuttals**

We have provided additional clarifications and experiments that **address the concerns raised by each reviewer**:

- Deeper discussions on the failure identification component from both conceptual and architectural perspectives, supported by quantitative results. (X6vo, PruS, 6UsG)
- Theoretical complexity analysis and evaluation of computational overhead. (X6vo)
- Comparative experiments on PIR variants with different retrieval and training processes. (PruS, 6UsG)
- Analysis of the variance in improvements across backbone models and datasets. (PruS)
- Comparison and discussion of recent related works. (PruS, 6UsG)

Thank you again for the insightful and constructive feedback. In the revised version, we will incorporate the discussed experiments, clarifications, and detailed case studies to further improve the quality and completeness of our work.

Sincerely,
The Authors of Submission 7958

---

### Decision · Program_Chairs · 2025-09-17

**Decision:**

Accept (poster)

**Comment:**

The authors proposed a novel post-training methodology, which is model-agnostic and designed to further enhance forecasting accuracy. The main advantage of this paper is the universal applicability of the proposed method, as it can be integrated with any existing forecasting model. Moreover, the additional results provided during the rebuttal period improve the validity of the proposed framework. One limitation of this study is that the performance improvements are sometimes limited for some forecasting models, especially when considering the additional computational resources required during the post-training phase. In conclusion, the contributions of the paper are clear, and the successful rebuttals have strengthened the authors' claims with more robust supporting evidence.